# Optimization of Glass Edge Sealing Process Using Microwaves for Fabrication of Vacuum Glazing

**Jae Kyung Kim [1], Young Shin Kim [1] and Euy Sik Jeon [1,2,*]**

[1]  Industrial Technology Research Institute, Kongju National University, Cheonan-daero, Seobuk-gu, Cheonan-si 31080, Chungcheongnam-do, Korea; kjk8431@gmail.com (J.K.K.); people9318@gmail.com (Y.S.K.)

[2]  Department of Mechanical Engineering, Graduate School, Kongju National University, Cheonan-daero, Seobuk-gu, Cheonan-si 31080, Chungcheongnam-do, Korea

*  Correspondence: osjun@kongju.ac.kr; Tel.: +82-41-521-9284

**Abstract:** Among the various methods used for glass edge sealing, this study uses microwaves to seal glass edges. Through basic experiments, the main process conditions for edge sealing of glass were derived, and the experimental plan and analysis were carried out using the Box-Behnken method of response-surface analysis based on 3 factors and 3 levels. The step height which influences sealing was set as a response variable. If the step height becomes too large, the glass can be damaged, and if the step height is too small, the edge sealing will not be performed. Accordingly, process optimization that edge sealing is possible while minimizing the step height was carried out. A predictable regression equation was derived for the step height of edge sealing and the main-effect analysis was performed for the step height. Using the response-optimization tool, we derived the optimum process condition that minimized the step height of the edge sealing and verified that it matched the error value of 4.1% compared with the target value of the step height measurement result confirmed through the verification experiment.

**Keywords:** microwave; glass edge sealing; DOE; process variable optimization

## 1. Introduction

A vacuum glass is a representative product wherein vacuum is used to improve the insulation performance. Various studies on fabricating vacuum glasses are currently being conducted, and glass edge sealing is a core process in the fabrication of vacuum glasses.

The edge sealing process is a technique whereby glass bonding (frit) is applied to the glass surface to bond two pieces of glass together. Numerous studies on this process have been conducted [1,2]. Edge sealing techniques using frits can be applied to various areas, from display technologies to home appliances and windows. However, the reduced strength due to the difference between the thermal expansion coefficients of the glass and glass bonding remains a problem [3].

Using hydrogen gas torches in the edge sealing process resolves the problem of the thermal expansion coefficient mismatch. However, the resulting sealed edges sag, making them inappropriate for panel fabrications [4,5]. Therefore, this study used microwaves to resolve this problem. Microwave is a strong energy source that has many practical applications in both industrial and commercial fields. It possesses excellent reproducibility and helps reduce the processing time. Moreover, the uniform heating process improves the quality of the final product.

When microwaves are used to sinter ceramics, the temperature of the material increases from the inside. Therefore, by combining microwaves with surface heating technologies, a highly uniform thermal energy distribution can be achieved. Furthermore, microwaves can be used to

heat specific regions, such as interfaces, by exploiting the interactions between the microwaves and the material. Thus, applying microwaves to the sealing process can result in a faster processing time, excellent performance, and allow for the manipulation of characteristics in the interfaces of composite materials [6–8].

In this paper, microwaves were used to seal glass edges. The levels of process variables for the glass edge sealing were determined through basic experiments, and the step height of the sealed edge was set as the characteristic value. Furthermore, the target step height of the sealed edge was determined through a liquid penetrant examination. Based on the process variables and the target step height, additional experiments were conducted for process optimization. For the optimization of edge sealing, a response-surface design was applied. ANOVA and regression equations were used to determine the sealing characteristics with respect to each process variable. Additionally, optimization tools were used to improve the process of deriving the target step height (characteristic value). Further experiments were carried out using the optimized process to test the validity of the process.

## 2. Experimental Setup

### 2.1. Equipment Setup

The microwave chamber for glass edge sealing was designed and constructed based on the electromagnetic wave distribution analysis using the HFSS program by ANSYS [6]. This experiment used the waveguide model WR-340 with a frequency range of 2.20–3.30 GHz, a voltage standing wave ratio of 1.25:1, and physical dimensions of 86.36 × 43.18 mm. The magnetron for microwave emission comprised six power sources with an output of 1 kW. The dimensions of the heating chamber were 400 (w) × 400 (d) × 192 (h) mm. Figure 1 shows the layout of the microwave chamber.

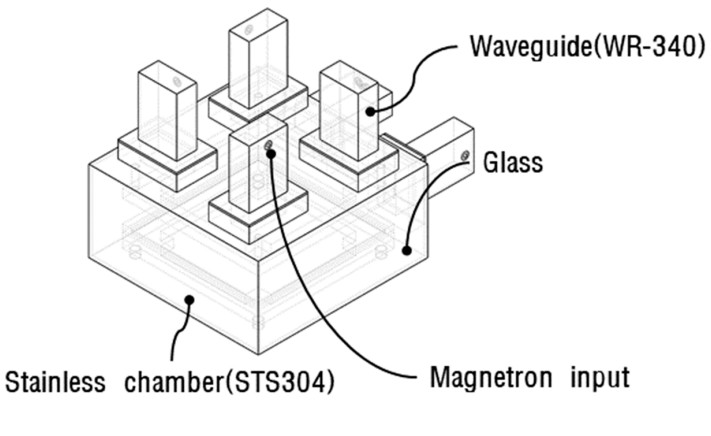

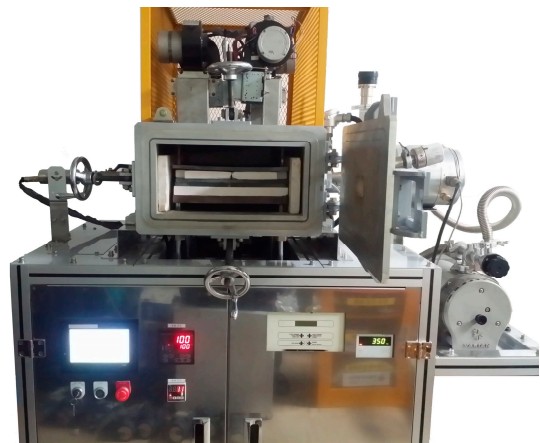

**Figure 1.** Schematic and real picture of microwave heating chamber.

A thin sheet of glass with a thickness of 0.5 mm was placed in between the glass and glass to keep the distance between the two soda lime glasses, each with a thickness of 5 mm. Generally, microwaves permeate through glass instead of reflecting from or heating the glass. However, manufacturers add impurities or additives so that the glass interacts with the microwaves. Therefore, when glass is heated using microwaves, the thermal energy is often concentrated resulting in thermal shocks and damage. To prevent this and enable efficient heat distribution, graphite plates were placed above and beneath the glass.

Figure 2a,b shows the schematics of the chamber for glass sealing and an image of the fabricated chamber, respectively.

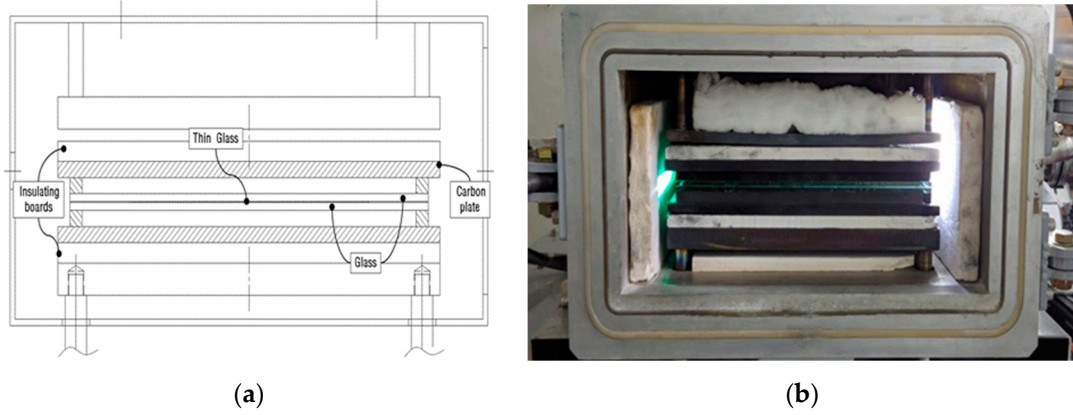

(**a**)                                        (**b**)

**Figure 2.** Schematic and image of experimental setup. (**a**) Schematics of the chamber for glass sealing; and (**b**) picture of the fabricated chamber.

### 2.2. Basic Experiment

The basic experiment was carried out to set the process and reaction variables. The factors that affected the quality of the sealed edge when the glass edges were sealed using microwaves were examined. Furthermore, different factors were derived in this stage to verify the characteristics of the sealed edge.

The microwave radiation and the glass edge sealing were carried out at the glass transition temperature in this experiment. The sealing result showed inadequately pressed parts on the edges, thus creating unsealed areas. A liquid penetrant examination was conducted to verify the sealing [9,10]. The liquid permeated through the glass to areas that were not sealed. Figure 3 shows an image of the liquid penetrant examination for the resulting edge sealing in the basic experiment.

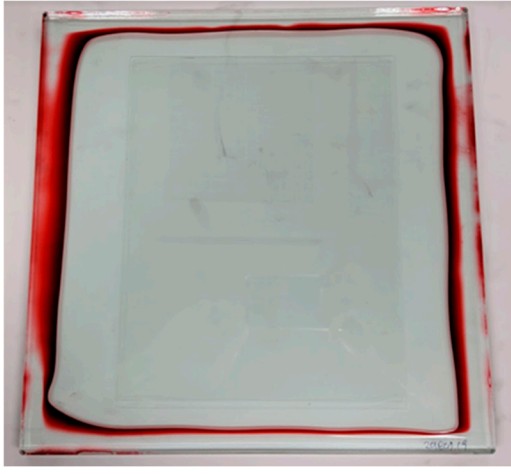

**Figure 3.** Liquid penetrant examination for the basic experiment.

If a high pressure was applied to the four edges, the probability of complete sealing increased. However, in that case, a large step height between the sealed edges and the center of the glass was generated. This may increase the surface stress of the glass after the experiment, which could lead to breakage. Thus, appropriate fabrication conditions were required such that the step height between the center of the glass and the edges were minimized while allowing for completely sealed edges.

To derive the minimal step height, five basic experiments were carried out with different step heights. In each experiment, the sealed glass was divided into four sections vertically and horizontally, resulting in 16 separate subsections. To measure the height of the separate regions, a laser displacement sensor (CD-33-30N-422; Optex, Ogoto, Japan) was used. Figure 4 shows the layout of the step height measuring system. Figure 5 shows the schematics of the step height measurement.

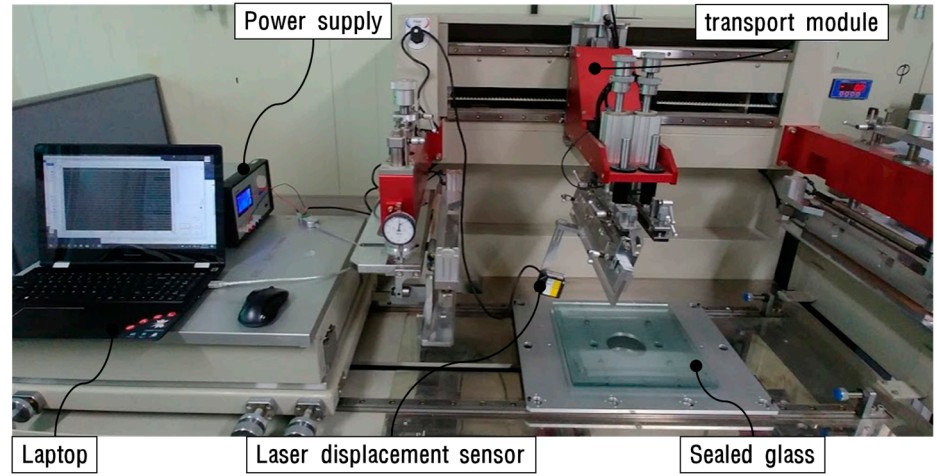

**Figure 4.** Step height measurement system layout.

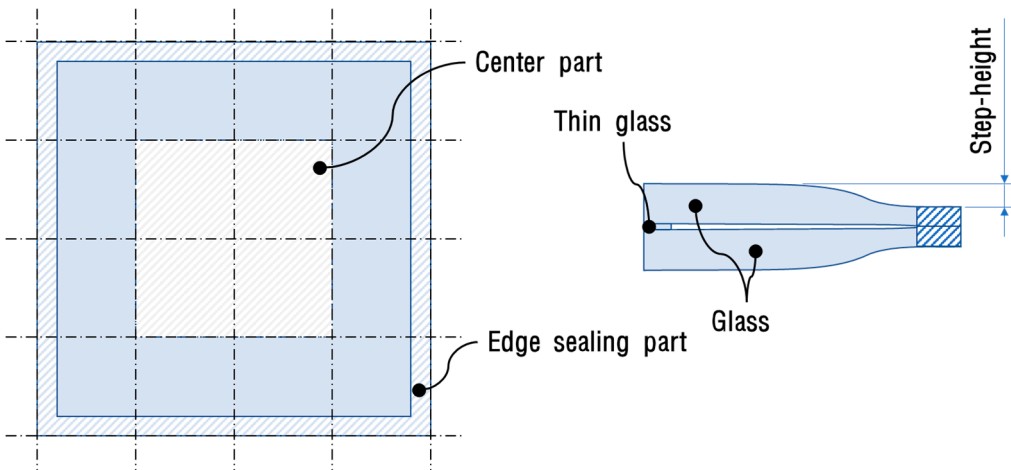

**Figure 5.** Step height measurement schematics.

After the glass height measurement, the measured glass sections were cut using the water jet technique. The exposed surfaces were put through liquid penetrant examination to ensure complete sealing. The sealed edges were further analyzed using Dino-Lite, which is a digital microscope. Figure 6 shows the cross-sections of the microwave sealed edges. Figure 6a shows an unsuccessfully sealed specimen, whereas Figure 6b shows a successfully sealed specimen.

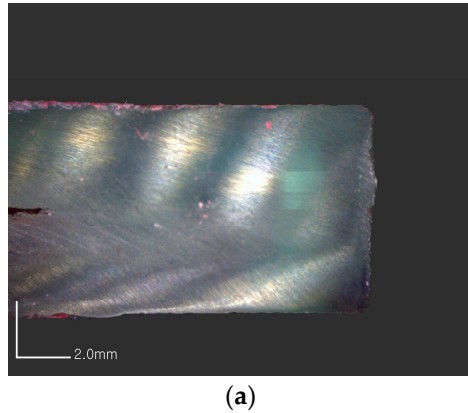
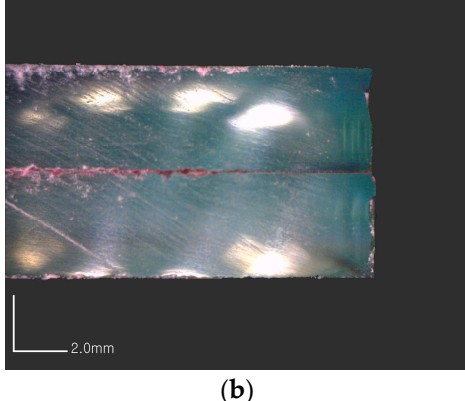

(**a**)　　　　　　　　　　　　　　　　　　　　(**b**)

**Figure 6.** Liquid penetrant examination of the cross-sections of sealed edges. (**a**) Cross-section of unsealed edge; and (**b**) cross-section of sealed edge.

The experimental results showed that if the step height was less than 0.66 mm, a significant portion of the sealed edge was not sealed. Even when the step height was the highest, i.e., 0.93 mm, there existed regions where the sealing was incomplete. This suggested that when the step height was 0.93 mm, sealing faults could occur in specific sections. Therefore, a minimum step height of 0.93 mm is required for complete sealing.

*2.3. Variable Setting*

The process and response variables were set by conducting basic experiments on the glass edge sealing using a microwave heating chamber. Among the various process variables used for glass edge sealing, the heating rate for the glass and the holding time for pressing the glasses together were set as the process variables. If the heating rate is too high, the glass could be damaged because of thermal shock [11–13], and if the heating rate is too low, the fabrication process could take too long. The sealing temperature was set approximately equal to the transition temperature of the glass. The holding time was set as the duration after the maximum temperature was reached, if this is too short the glass could break.

For glass edge sealing using microwaves, the thermal energy must be concentrated at the edges of the glass and a constant pressure must be maintained. The pressure was kept constant along all the edges owing to the design characteristics of the mechanical pressing device. Thus, in this research, three factors in three levels were identified in the basic experiment. The heating rate was set in the range of 6–8 °C/min, the sealing temperature was set in the range of 560–580 °C, and the holding time was set in the range of 30–50 min. Based on the response variable, it was determined in the basic experiment that the maximum step height corresponding to incomplete sealing was 0.93 mm. Therefore, considering the errors in the basic experiments, a nominal-is-best experiment was performed with a target step height of 0.98 mm in this study. Table 1 lists the conditions for the process and response variables.

**Table 1.** Conditions for the process and response variables.

| Factors | | Conditions | | |
|---|---|---|---|---|
| Process variables | Heating rate (°C/min) | 6 | 7 | 8 |
| | Sealing temperature (°C/min) | 560 | 570 | 580 |
| | Holding time (min) | 30 | 40 | 50 |
| Response variable | Step height (mm) | | 0.98 | |

### 3. Deriving Optimal Process Conditions Using Design of Experiments (DOE)

*3.1. Designing Experiments Using DOE*

Based on the process variables identified in the basic experiments, additional experiments were carried out to derive the optimal processing conditions wherein nominal-is-best characteristics were achieved by fabricating sealed edges with a uniform thickness. This experiment was designed using the Box–Behnken design, which is used for response-surface methodologies. The Box–Behnken design is a statistical method which compares the relationship of design variables (minimal 2) and the response level while also deriving the optimal process variables for the best response variable [14]. Table 2 lists the experimental conditions for each process variable. A total of 30 experiments were carried out.

*3.2. Analysis of Experimental Results*

The analysis based on the design condition tables were carried out to assess the significance of each factor and the significance of the interaction terms. Based on a 95% confidence, if the *p*-value was lower than 0.05, it was pooled as an error term and only the significant terms were verified. The glass sealing temperature and the holding time were found to affect the step height. Moreover, it was determined that the squared holding time, heating rate, and sealing temperature exhibited an interactive behavior. The $R^2$ value was the coefficient of determination and it showed the effectiveness of the model. If the $R^2$ value was close to 100%, it indicated that the model was representative of the observed values. An $R^2$ value of 66.80% was obtained using the regression equation based on the three process variables, indicating a significance within 5%. Table 3 lists the analyzed results. Equation (1) shows the obtained regression equation.

$$s_h = -107.8 + 12.46h_r + 0.1846s_t + 0.1942h_t - 0.002266h_t^2 - 0.02200h_r s_t \tag{1}$$

where:

$s_h$ = *Step height* (mm)
$h_r$ = *Heating rate* (/min)
$s_t$ = *Sealing temperature* ()
$h_t$ = *Holding time* (min)

**Table 2.** Experimental conditions for each process.

| Run Order | Heating Rate [°C/min] | Sealing Temp. [°C] | Holding Time [min] | Step-Height [mm] | Run Order | Heating Rate [°C/min] | Sealing Temp. [°C] | Holding Time [min] | Step-Height [mm] | Run Order | Heating Rate [°C/min] | Sealing Temp. [°C] | Holding Time [min] | Step-Height [mm] |
|---|---|---|---|---|---|---|---|---|---|---|---|---|---|---|
| 1 | 7 | 580 | 30 | 1.210 | 11 | 8 | 570 | 50 | 0.967 | 21 | 7 | 560 | 30 | 1.073 |
| 2 | 8 | 570 | 50 | 0.460 | 12 | 7 | 570 | 40 | 0.423 | 22 | 7 | 560 | 50 | 0.593 |
| 3 | 7 | 580 | 50 | 1.293 | 13 | 6 | 560 | 40 | 1.857 | 23 | 7 | 570 | 40 | 1.060 |
| 4 | 7 | 570 | 40 | 0.550 | 14 | 8 | 570 | 30 | 0.820 | 24 | 7 | 560 | 30 | 0.790 |
| 5 | 6 | 570 | 30 | 1.180 | 15 | 7 | 580 | 50 | 0.460 | 25 | 6 | 580 | 40 | 0.733 |
| 6 | 6 | 570 | 50 | 0.473 | 16 | 7 | 570 | 40 | 0.500 | 26 | 6 | 570 | 30 | 0.967 |
| 7 | 8 | 560 | 40 | 0.487 | 17 | 7 | 570 | 40 | 0.810 | 27 | 7 | 580 | 30 | 1.633 |
| 8 | 8 | 580 | 40 | 1.017 | 18 | 7 | 570 | 40 | 1.267 | 28 | 6 | 560 | 40 | 0.667 |
| 9 | 6 | 580 | 40 | 0.767 | 19 | 8 | 580 | 40 | 1.217 | 29 | 7 | 560 | 50 | 1.287 |
| 10 | 8 | 560 | 40 | 0.967 | 20 | 6 | 570 | 50 | 1.333 | 30 | 8 | 570 | 30 | 0.693 |

**Table 3.** ANOVA results.

| Term | Coef | SE Coef | $T-$Value | $P-$Value | VIF |
|------|------|---------|-----------|-----------|-----|
| Constant | 1.0393 | 0.0624 | 16.66 | 0.000 | |
| $h_r$ | −0.0785 | 0.0584 | −1.35 | 0.191 | 1.00 |
| $s_t$ | 0.3058 | 0.0584 | 5.24 | 0.000 | 1.00 |
| $h_t$ | 0.1290 | 0.0584 | 2.21 | 0.037 | 1.00 |
| $h_t^2$ | −0.2266 | 0.0854 | −2.65 | 0.014 | 1.00 |
| $h_t \times s_t$ | −0.2200 | 0.0825 | −2.67 | 0.014 | 1.00 |

Figures 7 and 8 show the contour and surface plots of the step height based on the process variables. The step height showed a curvature effect depending on the holding time. It was also found that the step height of the sealed edge was less affected by the heating rate, and that the step height was proportional to the temperature of the sealed edge.

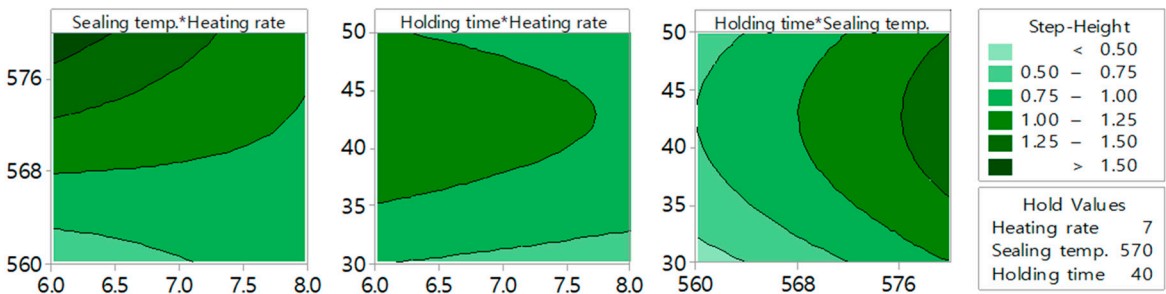

**Figure 7.** Contour plot of the step height based on process variables.

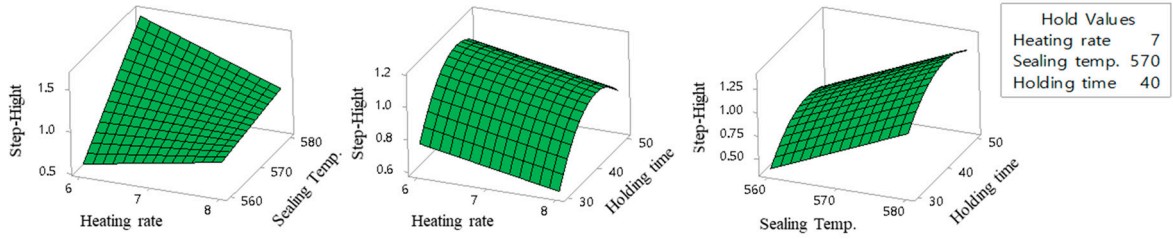

**Figure 8.** Surface plot of the step height based on the process variables.

## 4. Fabrication Optimization

### 4.1. Deriving Optimal Fabrication Conditions

To satisfy the minimal step height of 0.98 mm, as previously determined in the basic experiment, optimal fabrication conditions were derived using response-optimization tools. The optimal values of the heating rate, sealing temperature, and holding time were found to be 6.13 °C/min, 573.13 °C, and 31.30 min, respectively. Figure 9 shows the graph of the optimized response, which was used to predict the step height of the sealed edge based on the process variables.

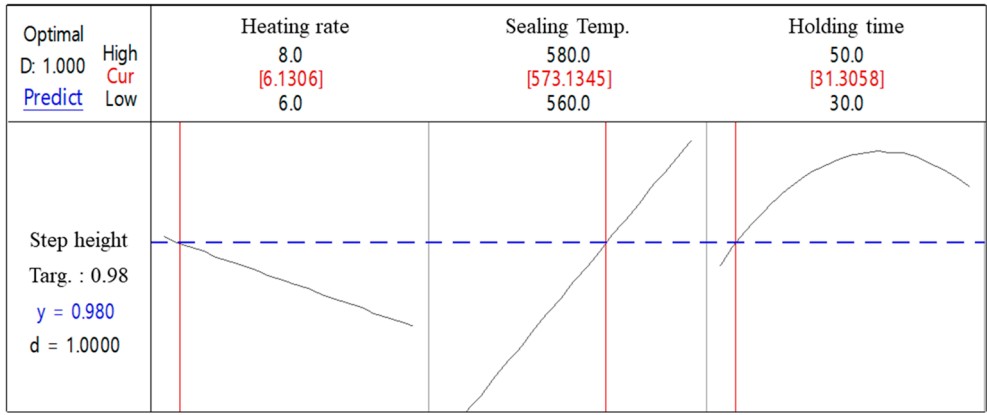

**Figure 9.** Optimized response graph.

### 4.2. Verification of the Optimized Process Conditions

To verify the optimized process conditions derived in the experimental design section, verification experiments were carried out. In this experiment, the heating rate of the microwave chamber was set to 6 °C/min, the sealing temperature was set to 573 °C, and the holding time was set to 31 min. Due to equipment limitations, digits after the decimal point were omitted.

Laser sensors were used to measure the step height of the glass sample created under the derived optimized fabrication conditions. The step height was found to be 0.94 mm, which was 4.1% lower than the target step height. Furthermore, a liquid penetrant examination was applied to ensure that the edges were sealed completely. Figure 10 shows the glass sample created under the derived optimized fabrication conditions.

The sealed glass was cut through water jet processing and the sealed cross-section was observed using a digital microscope. Figure 11 shows the observation results, it was confirmed that all glass edges were completely sealed.

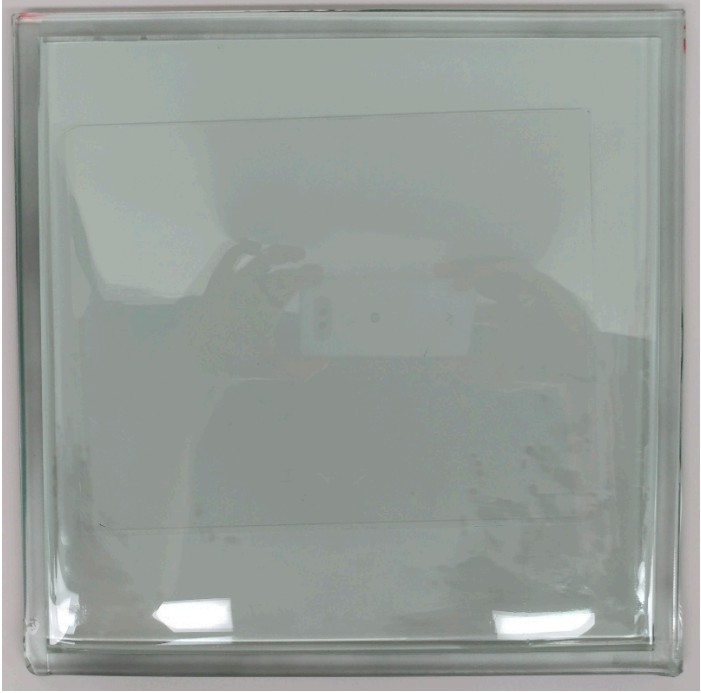

**Figure 10.** Glass sample sealed using the optimized fabrication conditions.

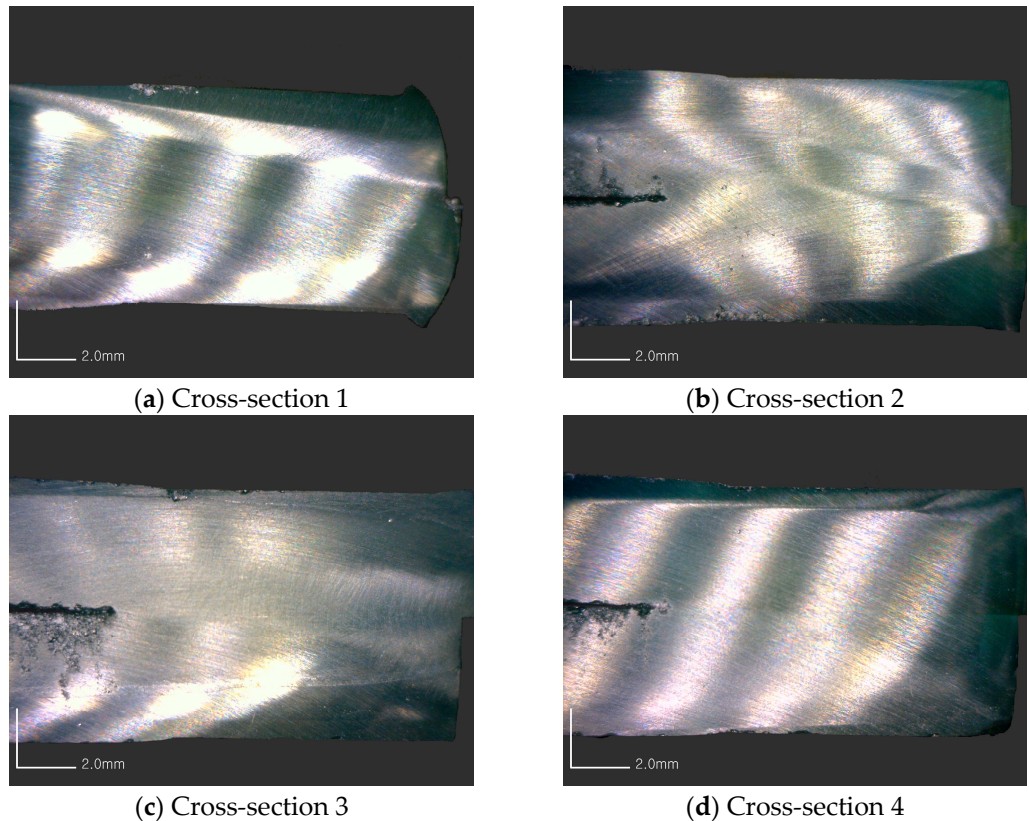

(**a**) Cross-section 1　　　　　　　　　　　　　　(**b**) Cross-section 2

(**c**) Cross-section 3　　　　　　　　　　　　　　(**d**) Cross-section 4

**Figure 11.** Observation results of cross-section of sealed glass.

## 5. Conclusions

When microwaves are used to seal the glass edges, the sealed edges are pressed because of the various process variables. Therefore, process conditions were derived to optimize the edge sealing based on the pressed edges.

The initial process conditions under which the edges were sealed using microwaves were determined through basic experiments. Furthermore, to analyze the sealing characteristics based on the process conditions, the step heights were measured and water jets were used to cut the sealed glass. A liquid penetrant examination was then applied to obtain a complete sealing, based on which the target step height of the pressed edge was set. To analyze the correlations between the three factors and the three levels of the process variables established through the basic experiment and the pressing of the sealed edge, as well as to derive the optimal process conditions for edge sealing, additional experiments were performed using the response-surface methodology, which was a DOE method. The regression equation was derived by analyzing the experimental result, and the heating rate of 6 °C/min, the sealing temperature 573 °C, and the holding time 31 min of the optimum process conditions were derived reflecting the nominal-is-best characteristics. Additional experiments were conducted based on derived optimal process conditions. By comparing the predicted value of the optimum condition with the experimental value and confirming the error of 4.1%, the validity of the optimum process condition and the regression equation was verified.

It is expected that the glass edge sealing process using microwaves under the optimal conditions derived in this study can be applied to the production of vacuum glasses.

**Author Contributions:** J.K.K., Y.S.K. and E.S.J. conceived and designed the experiments; J.K.K. performed the experiments; J.K.K. and Y.S.K. analyzed the data; J.K.K., Y.S.K. and E.S.J. contributed reagents/materials/analysis tools; J.K.K., Y.S.K. and E.S.J. wrote the paper.

**Funding:** This research was supported by Basic Science Research Program through the National Research Foundation of Korea (NRF) funded by the Ministry of Science and ICT (NRF-2017R1A2B4009929, 2018R1A2B6004348).

**Conflicts of Interest:** The authors declare no conflict of interest.

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
