# Peer review of "Optimization of Glass Edge Sealing Process Using Microwaves for Fabrication of Vacuum Glazing"

_applsci, doi:10.3390/app9050874_

Round 1

Reviewer 1 Report

I think the results must be improved by showing the effectiveness of the process (by perfoming Scanning electron Microscopy or others microscope observations).

The cross-section of the sealed edges must be clearly visible by microscopic images.

Author Response

I am truly grateful for your comments.

Point 1: I think the results must be improved by showing the effectiveness of the process (by performing Scanning electron Microscopy or others microscope observations).

Response 1: The process is in progress to prepare this material. Water jet cutting and section polishing work are necessary for taking the cross-section shape. This process needs more time about two weeks. After extending the period, we will submit the modifications reflecting the cross-section shape after completion of the process.

Point 2: The cross-section of the sealed edges must be clearly visible by microscopic images.

Response 2: It was revised.

Reviewer 2 Report

1.     Page 2, the sentences of the last paragraph of introduction section are very similar to your abstract, some sentence is exactly the same. Please delete or modify this part.

2.      Page 2, figure 1, if possible, please also add a real picture to compare with your schematic of chamber.

3.     Page 3, line 91, the sentence “creating a significant amount of surface stress, which can lead to damaged products.” There’s grammar error here, please modify this sentence.

4.     Page 4, figure 6, please add scale bar for figures.

5.     Page 5, line 130, the unit for heating rate is written incorrectly.  At line 186,  has the same typo.

6.     Pager 9, your conclusion is too general, please give the specific result after finishing the fabrication optimization.

Author Response

I am truly grateful for your comments.

Point 1: Page 2, the sentences of the last paragraph of introduction section are very similar to your abstract, some sentence is exactly the same. Please delete or modify this part.

Response 1: It was revised. We corrected the abstract.

Among the various methods used for glass edge sealing, this study uses microwaves to seal glass edges. Through basic experiments, the main process conditions for edge sealing of glass were derived, and the experimental plan and analysis were carried out using the Box-Behnken method of response-surface analysis based on 3 factors and 3 levels. As the response variable, we set whether or not to seal with reference to the step due to pressure at the edge of glass and optimize the process to minimize the step height to prevent breakage of glass due to step height. A predictable regression equation was derived for the step height of edge sealing and performed the main-effect analysis was performed for the step height. Using the response-optimization tool, we derived the optimum process condition that minimizes the step height of the edge sealing and verify that it matches the error value of 4.1% compared with the target value of the step-height measurement result confirmed through the verification experiment.

Point 2: Page 2, figure 1, if possible, please also add a real picture to compare with your schematic of chamber.

Response 2: We added the real picture.

Point 3: Page 3, line 91, the sentence “creating a significant amount of surface stress, which can lead to damaged products.” There’s grammar error here, please modify this sentence.

Response 3: It was revised.

However, in that case, a large step height between the sealed edges and the centre of the glass is greatly generated. This may increase the surface stress of the glass after the experiment, which may lead to breakage.

Point 4: Page 4, figure 6, please add scale bar for figures.

Response 4: It was revised.

Point 5: Page 5, line 130, the unit for heating rate is written incorrectly.  At line 186,  has the same type.

Response 5: It was revised.

Point 6: Pager 9, your conclusion is too general, please give the specific result after finishing the fabrication optimization.

Response 6: It was revised.

When microwaves are used to seal the glass edges, the sealed edges are pressed because of the various process variables. Therefore, process conditions were derived to optimize the edge sealing based on the pressed edges.

The initial process conditions under which the edges are sealed using microwaves were determined through basic experiments. Furthermore, to analyze the sealing characteristics based on the process conditions, the step heights were measured, and water jets were used to cut the sealed glass. A liquid penetrant examination was then applied to obtain a complete sealing, based on which the target step height of the pressed edge was set. To analyze the correlations between the three factors and the three levels of the process variables established through the basic experiment and the pressing of the sealed edge and to derive the optimal process conditions for edge sealing, additional experiments were performed using the response surface methodology, which is a DOE method. The regression equation was derived by analyzing the experimental result, and the heating rate of 6/min, the sealing temperature 573, and the holding time 31min of the optimum process conditions were derived reflecting the nominal-is-best characteristics. Additional experiments were conducted based on derived optimal process conditions. By comparing the predicted value of the optimum condition with the experimental value and confirming the error of 4.1%, the validity of the optimum process condition and the regression equation was verified.

 It is expected that the glass edge sealing process using microwaves under the optimal conditions derived in this study can be applied to the production of vacuum glasses.

Round 2

Reviewer 1 Report

I suggest the pubblication in present form.